# Improved Regret Analysis for Variance-Adaptive Linear Bandits and Horizon-Free Linear Mixture MDPs

**Yeoneung Kim**[*]
Gachon University
yeoneung@gachon.ac.kr

**Insoon Yang**
Seoul National University
insoonyang@snu.ac.kr

**Kwang-Sung Jun**
University of Arizona
kjun@cs.arizona.edu

## Abstract

In online learning problems, exploiting low variance plays an important role in obtaining tight performance guarantees yet is challenging because variances are often not known a priori. Recently, considerable progress has been made by Zhang et al. (2021) where they obtain a variance-adaptive regret bound for linear bandits without knowledge of the variances and a horizon-free regret bound for linear mixture Markov decision processes (MDPs). In this paper, we present novel analyses that improve their regret bounds significantly. For linear bandits, we achieve $\tilde{O}(\min\{d\sqrt{K}, d^{1.5}\sqrt{\sum_{k=1}^{K}\sigma_k^2}\} + d^2)$ where $d$ is the dimension of the features, $K$ is the time horizon, and $\sigma_k^2$ is the noise variance at time step $k$, and $\tilde{O}$ ignores polylogarithmic dependence, which is a factor of $d^3$ improvement. For linear mixture MDPs with the assumption of maximum cumulative reward in an episode being in $[0, 1]$, we achieve a horizon-free regret bound of $\tilde{O}(d\sqrt{K} + d^2)$ where $d$ is the number of base models and $K$ is the number of episodes. This is a factor of $d^{3.5}$ improvement in the leading term and $d^7$ in the lower order term. Our analysis critically relies on a novel peeling-based regret analysis that leverages the elliptical potential 'count' lemma.

## 1 Introduction

In online learning, variance often plays an important role in achieving low regret bounds. For example, for the prediction with expert advice problem, Hazan and Kale [11] proposed an algorithm that achieves a regret bound of $O(\sqrt{\text{VAR}_K})$ where $\text{VAR}_K$ is a suitably-defined variance of the loss function up to time step $K$, without knowing $\text{VAR}_K$ ahead of time. The implication is that when the given sequence of loss functions has a small variance, one can perform much better than the previously known regret bound $O(\sqrt{K})$. For multi-armed bandits, Audibert et al. [2] proposed an algorithm that achieves regret bounds that depends on the variances of the arms, which means that, again, the regret bound becomes smaller as the variances become smaller.

It is thus natural to obtain similar variance-adaptive bounds for other problems. For example, in $d$-dimensional stochastic contextual bandit problems, the optimal worst-case regret bound is $\tilde{O}(\sigma d\sqrt{K})$ where $\tilde{O}$ hides polylogarithmic dependencies and $\sigma^2$ is a uniform upper bound on the noise variance. Following the developments in other online learning problems, it is natural to ask if we can develop a similar variance-adaptive regret bound. The recent work by Zhang et al. [31] has provided an affirmative answer. Their algorithm called VOFUL achieves a regret bound of $\tilde{O}(d^{4.5}\sqrt{\sum_{k=1}^{K}\sigma_k^2} + d^5)$ where $\sigma_k^2$ is the (unknown) noise variance at time step $k$. This implies that, indeed, it is possible

---

[*]Work done while at Seoul National University.

36th Conference on Neural Information Processing Systems (NeurIPS 2022).

to adapt to the variance and suffer a much lower regret. Furthermore, they show that a similar variance-adaptive analysis can be used to solve linear mixture Markov decision processes (MDPs) with the *unit cumulative rewards* assumption :

$$\sum_h r_h^k \in [0, 1], \forall k \tag{1}$$

where $r_h^k$ is the reward received at episode $k$ and horizon $h$. They show a regret bound of $\tilde{O}(d^{4.5}\sqrt{K} + d^9)$, which does not depend on the planning horizon length $H$ up to polylogarithmic factors. We elaborate more on the linear bandit and linear mixture MDP problems in Section 2.

However, the regret rates of these problems have a large gap between the known lower and the upper bounds. For example, in linear bandits, it is well-known that the regret bound has to be $\Omega(d\sqrt{K})$ [6], which rejects the possibility of obtaining $o(d\sqrt{\sum_{k=1}^K \sigma_k^2})$, yet the best upper bound obtained so far is $O(d^{4.5}\sqrt{\sum_{k=1}^K \sigma_k^2})$. Thus, the gap is a factor of $d^{3.5}$, which is quite large.

In this paper, we reduce such gaps significantly by obtaining much tighter regret upper bounds. Specifically, we show that a slight variation of VOFUL [31] for linear bandits has a regret bound of $\tilde{O}(\min\{d\sqrt{K}, d^{1.5}\sqrt{\sum_{k=1}^K \sigma_k^2}\})$ without knowledge of the variances. This reduces the gap between the upper and lower bounds to only $\sqrt{d}$ for the leading term in the regret. Furthermore, we employ a similar technique to show that the algorithm VARLin [31] for linear mixture MDPs with unit cumulative rewards has a regret bound of $\tilde{O}(d\sqrt{K} + d^2)$. At the heart of our analysis is a direct peeling of the instantaneous regret terms using an elliptical potential 'count' lemma (EPC). EPC bounds, given $q > 0$, how many times $\|x_k\|^2_{V_{k-1}^{-1}} \geq q$ happens from time $k = 1$ to $\infty$ where $V_{k-1} = \sum_{s=1}^{k-1} x_s x_s^\top$. Our lemma is an improved and generalized version of [16, Exercise 19.3], which was originally used for improving the regret bound of linear bandit algorithms. We provide the proofs of our main results for linear bandits and linear mixture MDPs in Section 3 and Section 4 respectively. Finally, we conclude the paper with exciting future directions.

**Related work.** There are numerous works on linear bandit problems such as [6, 4, 1, 17] where the information of variance is not used. On the other hand, variance can be exploited to obtain better regret [2]. Recently, works by [31, 33] proposed ways to infuse the variance information in the regret analysis which improves the standard regret bound. Reinforcement learning with linear function approximation has been widely studied to develop efficient learning methods that work for large state-action space [27, 26, 13, 8, 14, 23, 24, 25, 29, 18, 15, 7, 21, 10, 9, 28]. To our knowledge, all aforementioned works derived a regret bound that depends on the planning horizon $H$ polynomially. It was Zhang et al. [31] who first remove the polynomial dependence of $H$ in the linear mixture MDP problem, achieving a bound of $\tilde{O}(d^{4.5}\sqrt{K} + d^9)$. In contrast, our analysis shows that their algorithm in fact achieves significantly better bound of $\tilde{O}(d\sqrt{K} + d^2)$. Note that these results assume the unit cumulative rewards assumption (1) and time-homogeneous transition models. In a similar setup where $r_{h,k} \in [0, 1]$ with time-inhomogeneous transition models, Zhou et al. [33] achieve the regret $\tilde{O}(\sqrt{d^2 H + dH^3}\sqrt{HK} + d^2 H^3 + d^3 H^2)$ and show a lower bound of $\tilde{\Omega}(dH^{3/2}\sqrt{K})$. These problem setups are incompatible to the setup of [31] and ours.

## 2 Problem Definition

**Notations.** We denote $d$-dimensional $\ell_2$ ball by $\mathbb{B}_2^d(R) := \{x \in \mathbb{R}^d : \|x\|_2 \leq R\}$ and define $\mathbb{B}_1^d(R)$ similarly for the $\ell_1$ ball. Let $[N] := \{1, 2, \dots, N\}$ for $N \in \mathbb{N}$. Given $\ell \in \mathbb{R}$ and $x \in \mathbb{R}$, we define the clipping operator as follows (take $0/0 = 0$):

$$\overline{(x)}_\ell := \min\left\{|x|, 2^{-\ell}\right\} \cdot \frac{x}{|x|} . \tag{2}$$

**Linear bandits.** The linear bandit problem has the following protocol. At time step $k$, the learner observes an arm set $\mathcal{X}_k \subseteq \mathbb{B}_2^d(1)$, chooses an arm $x_k \in \mathcal{X}_k$, pulls it. The learner then receives a stochastic reward $r_k = x_k^\top \theta^* + \epsilon_k$ where $\theta^* \in \mathbb{B}_2^d(1)$ is an unknown parameter and $\epsilon_k$ is a zero-mean stochastic noise. Following [31], we assume that $(i)$ $\forall k \in [K]$, $|r_k| \leq [-\frac{1}{2}, \frac{1}{2}]$ almost surely, $(ii)$ $\mathbb{E}[\epsilon_k|\mathcal{F}_k] = 0$ where $\mathcal{F}_k = \sigma(x_1, \epsilon_1, ..., x_{k-1}, \epsilon_{k-1}, x_k)$, and $(iii)$ $\mathbb{E}[\epsilon_k^2|\mathcal{F}_k] = \sigma_k^2$ .

Note that the bound on $|r_k|$ implies that $|\epsilon_k| \leq 1$ almost surely. Our goal is to minimize the regret $\mathcal{R}^K = \sum_{k=1}^K \max_{x \in \mathcal{X}_k} x^\top \theta^* - x_k^\top \theta^*$.

**Linear mixture MDPs.** We consider an episodic Markov Decision Process (MDP) with a tuple $(\mathcal{S}, \mathcal{A}, r(s,a), P(s'|s,a), K, H)$ where $\mathcal{S}$ is the state space, $\mathcal{A}$ is the action space, $r : \mathcal{S} \times \mathcal{A} \to [0,1]$ is the reward function, $P(s'|s,a)$ is the transition probability, $K$ is the number of episodes, and $H$ is the planning horizon. A policy is defined as $\pi = \{\pi_h : \mathcal{S} \to \mathcal{D}(\mathcal{A})\}_{h=1}^H$ where $\mathcal{D}(\mathcal{A})$ is a set of all distributions over $\mathcal{A}$. For each episode $k \in [K]$, the learner chooses a policy $\pi^k$, and then the environment executes $\pi^k$ on the MDP by successively following $a_h^k \sim \pi_h^k(s_h^k)$ and $s_{h+1}^k \sim P(\cdot|s_h^k, a_h^k)$. Then, the learner observes the rewards $\{r_h^k \in [0,1]\}_{k,h}$ and moves onto the next episode. The key modeling assumption of linear mixture MDPs is that the transition probability $P$ is a linear combination of a known set of models $\{P^i\}$, namely, $P = \sum_{i=1}^d \theta_i^* P^i$ where $\theta^* \in \mathbb{B}_1^d(1)$ is an unknown parameter. We follow [31] and make the following assumptions:

- The reward at each time step $h$ and episode $k$ is $r_h^k = r(s_h^k, a_h^k)$ for some known function $r : \mathcal{S} \times \mathcal{A} \to [0,1]$.
- Unit cumulative rewards: $\sum_{h=1}^H r_h^k \in [0,1]$ for any policy $\pi^k$.

For a policy $\pi$, $V_h^\pi(s) := \max_{a \in \mathcal{A}} Q_h^\pi(s,a)$ where $Q_h^\pi(s,a) = r(s,a) + \mathbb{E}_{s' \sim P(\cdot|s,a)} V_{h+1}^\pi(s')$ and $V_{H+1}^\pi(s) := 0$. Denoting $V^\pi(s_1) = V_1^\pi(s_1)$ and $V^*(s_1) = V^{\pi^*}(s_1)$, our goal is to minimize the regret

$$\mathcal{R}^K = \sum_{k=1}^K V^*(s_1^k) - V^k(s_1^k) \,.$$

## 3 Variance-Adaptive Linear Bandits

In this section, we show that VOFUL of Zhang et al. [31] has a tighter regret bound than what was reported in their work. Our version of VOFUL, which we call VOFUL2, has a slightly different confidence set for ease of exposition. Specifically, we use a confidence set that works for every $\mu \in \mathbb{B}_2^d(2)$ rather than over an $\epsilon$-net of $\mathbb{B}_2^d(2)$ (but we do use an $\epsilon$-net for the proof of the confidence set).

The full pseudocode can be found in Algorithm 1. VOFUL2 follows the standard optimism-based arm selection [3, 6, 1]. Let $\epsilon_s(\theta) := r_s - x_s^\top \theta$ and $\epsilon_s^2(\theta) := (\epsilon_s(\theta))^2$. With $L$ and $\iota$ defined in Algorithm 1, we define our confidence set after $k$ time steps as

$$\Theta_k := \cap_{\ell=1}^L \Theta_k^\ell \tag{3}$$

where

$$\Theta_k^\ell := \left\{ \theta \in \mathbb{B}_2^d(1) : \left| \sum_{s=1}^k \overline{(x_s^\top \mu)}_\ell \epsilon_s(\theta) \right| \leq \sqrt{\sum_{s=1}^k \overline{(x_s^\top \mu)}_\ell^2 \epsilon_s^2(\theta) \iota} + 2^{-\ell} \iota, \forall \mu \in \mathbb{B}_2^d(2) \right\}$$

and the clipping operator $\overline{(z)}_\ell$ is defined in (2). The role of clipping is two-fold: (i) it allows us to factor out $\sum_{s=1} \epsilon_s^2(\theta)$ by $\sum_s \overline{(x_s^\top \mu)}_\ell^2 \epsilon_s^2(\theta) \leq (2^{-\ell})^2 \sum_{s=1} \epsilon_s^2(\theta)$ and (ii) the lower order term is reduced to the order of $2^{-\ell}$. Both properties are critical in obtaining variance-adaptive regret bounds as discussed in [31]. The true parameter is contained in our confidence set with high probability as follows.

**Lemma 1.** *(Confidence set) Let L, $\iota$, and $\delta$ be given as those in Algorithm 1. Then,*

$$\mathbb{P}(\mathcal{E}_1 := \{\forall k \in [K], \theta^* \in \Theta_k\}) \geq 1 - \delta \,.$$

In fact, in our algorithm, we use the confidence set of $\cap_{s=1}^{k-1} \Theta_s$ at time step $k$ for a technical reason. VOFUL2 has the following regret bound.

**Theorem 1.** *VOFUL2 satisfies, with probability at least $1 - 2\delta$,*

$$\mathcal{R}^K = \tilde{O}\left( d^{1.5} \sqrt{\sum_{k=1}^K \sigma_k^2 \ln(1/\delta)} + d^2 \ln(1/\delta) \right)$$

---
**Algorithm 1** VOFUL2
---
1: **Initialize:** $L = 1 \vee \lfloor \log_2(K) \rfloor$ where $\iota = 128 \ln((12K2^L)^{d+2}/\delta)$ and $\delta \le e^{-1}$.
2: **for** $k = 1, 2, \ldots, K$ **do**
3:   Observe a decision set $\mathcal{X}_k \subseteq \mathbb{B}_2^d(1)$.
4:   Compute the optimistic arm as following: $x_k = \arg\max_{x \in \mathcal{X}_k} \max_{\theta \in \cap_{s=1}^{k-1} \Theta_s} x^\top \theta$ where $\Theta_s$ is
    defined in (3).
5:   Receive a reward $r_k$.
6: **end for**
---

where $\tilde{O}$ hides poly-logarithmic dependence on $\{d, K, \sum_{k=1}^{K} \sigma_k^2, \ln(1/\delta)\}$.

Note that one can also show that VOFUL2 can be slightly modified to achieve the regret bound of $\tilde{O}\left(\min\left\{d\sqrt{K\ln(1/\delta)}, d^{1.5}\sqrt{\sum_{k=1}^{K}\sigma_k^2 \ln(1/\delta)}\right\} + d^2 \ln(1/\delta)\right)$, thus being no worse than OFUL. We postpone the proof of this to Section A.4 to avoid clutter.

**Properties of the confidence sets and implications on the regret.** Before presenting the proof of Theorem 1, we provide some key properties of our confidence set (Lemma 3) and the intuition behind our regret bound. First, let us describe a few preliminaries. Define

$$W_{\ell,k-1}(\mu) := 2^{-\ell}I + \sum_{s=1}^{k-1}\left(1 \wedge \frac{2^{-\ell}}{|x_s^\top \mu|}\right)x_s x_s^\top.$$

Let $\theta_k$ be the maximizer of the optimization problem at line 4 of Algorithm 1 and define $\mu_k = \theta_k - \theta^*$. For brevity, we use a shorthand of

$$W_{\ell,k-1} := W_{\ell,k-1}(\mu_k) = 2^{-\ell}I + \sum_{s=1}^{k-1}\left(1 \wedge \frac{2^{-\ell}}{|x_s^\top \mu_k|}\right)x_s x_s^\top.$$

Finally, we need to define the following event regarding the concentration of the empirical variance around the true variance:

$$\mathcal{E}_2 := \left\{\forall k \in [K], \sum_{s=1}^{k}\epsilon_s^2(\theta^*) \le \sum_{s=1}^{k}8\sigma_s^2 + 4\log\left(\frac{4K(\log_2(K)+2)}{\delta}\right)\right\},$$

which is true with high probability as follows.

**Lemma 2.** *We have* $\mathbb{P}(\mathcal{E}_2) \ge 1 - \delta$

*Proof.* The proof is a direct consequence of Lemma 13 in our appendix. $\qquad\square$

Let $\ell_k$ be the integer $\ell$ such that $x_k^\top \mu_k \in (2 \cdot 2^{-\ell}, 2 \cdot 2^{-\ell+1}]$ and define $A_k := \sum_{s=1}^{k}\sigma_s^2$. Lemma 3 below states the properties of our confidence set.

**Lemma 3.** *Suppose the events $\mathcal{E}_1$ and $\mathcal{E}_2$ are true. Then, for any $k$ with $\ell_k = \ell$,*

(i) *For some absolute constant $c_1$,*
$$\|\mu_k\|_{W_{\ell,k-1}}^2 \le 2^{-\ell}\sqrt{128 A_{k-1}\iota} + 11 \cdot 2^{-\ell}\iota \le c_1 2^{-\ell}(\sqrt{A_{k-1}\iota} + \iota),$$

(ii) *There exists an absolute constant $c_2$ such that $x_k^\top \mu_k \le c_2 \|x_k\|_{W_{\ell,k-1}^{-1}}^2 \left(\sqrt{A_{k-1}\iota} + \iota\right)$.*

The key difference between Lemma 3 and the results of Zhang et al. [31] is that we use the norm notations, although the norm involves a rather complicated matrix $W_{\ell,k-1}$. This opens up possibilities of analyzing the regret of VOFUL2 with existing tools such as applying Cauchy-Schwarz inequality and the elliptical potential lemma [1, 5, 16]. In particular, Lemma 3(ii) seems useful because if we had such a result with $W_{\ell,k-1}$ replaced by $V_{k-1} = \lambda I + \sum_{s=1}^{k-1} x_s x_s^\top$, then we would have, ignoring the additive term $\iota$,

$$x_k^\top \mu_k \le \|x_k\|_{V_{k-1}^{-1}}^2 \sqrt{\sum_{s=1}^{k-1}\sigma_s^2 \iota}.$$

Together with the optimism and the standard elliptical potential lemma (see Section 3.1 for details), this leads to

$$\mathcal{R}^K \leq \sum_{k=1}^{K} x_k^\top \mu_k \leq c_2 \sum_{k=1}^{K} \|x_k\|_{V_{k-1}^{-1}}^2 \sqrt{\sum_{s=1}^{k-1} \sigma_s^2 \iota} \leq c_2 \cdot O(d \log(T/d)) \cdot \sqrt{\sum_{s=1}^{K} \sigma_s^2 \iota} \ .$$

Since $\iota$ is linear in $d$, we would get the regret bounded by the order of $d^{1.5} \sqrt{\sum_{k=1}^{K} \sigma_k^2}$, roughly speaking. However, the discrepancy between $W_{\ell,k-1}$ and $V_{k-1}$ is not trivial to resolve, especially due to the fact that Lemma 3(ii) has $\mu_k$ on both left and the right hand side. That is, $\mu_k$ is the key quantity that we need to understand, but we are bounding $x_k \mu_k$ as a function of $\mu_k$. The novelty of our analysis of regret is exactly at relating $W_{\ell,k-1}$ to $V_{k-1}$ via a novel peeling-based analysis, which we present below.

## 3.1 Proof of Theorem 1

Throughout the proof, we condition $\mathcal{E}_1$ and $\mathcal{E}_2$ where each one is true with probability at least $1 - \delta$, as shown in Lemma 1 and 2 respectively. For our regret analysis, it is critical to use Lemma 4 below, which we call the elliptical 'count' lemma. This lemma is a generalization of Lattimore and Szepesvári [16, Exercise 19.3], which was originally used therein to improve the dependence of the range of the expected rewards in the regret bound. Similar lemmas have been used in parallel studies [12, 22]. In particular, He et al. [12] employ a lemma similar to elliptical potential count and peeling technique for the regret analysis for the linear MDP as well, which we compare in detail in Section C due to space constraint. We remark that a similar strategy appears in disguise in Russo and Van Roy [20, Proposition 3] as well.

**Lemma 4.** *(Elliptical potential count) Let $x_1, \ldots, x_k \in \mathbb{R}^d$ be a sequence of vectors with $\|x_s\|_2 \leq X$ for all $s \in [k]$. Let $V_k = \tau I + \sum_{s=1}^{k} x_s x_s^\top$ for some $\tau > 0$. Let $J \subseteq [k]$ be the set of indices where $\|x_s\|_{V_{s-1}^{-1}}^2 \geq q$. Then,*

$$|J| \leq \frac{2}{\ln(1+q)} d \ln \left(1 + \frac{2/e}{\ln(1+q)} \frac{X^2}{\tau}\right) .$$

As the name explains, the lemma above bounds how many times $\|x_s\|_{V_{s-1}^{-1}}^2$ can go above a given value $q > 0$, which is different from existing elliptical potential lemmas that bound the sum of $\|x_s\|_{V_{s-1}^{-1}}^2$. Let $\theta_k$ be the $\theta$ that maximizes the optimization problem at line 4 of Algorithm 1. We start by the usual optimism-based bounds: due to $\mathcal{E}_1$, we have

$$\mathcal{R}^K = \sum_{k=1}^{K} \left(\max_{x \in \mathcal{X}_k} (x^\top \theta^* - x_k^\top \theta^*)\right) \leq \sum_{k=1}^{K} \left(\max_{x \in \mathcal{X}_k, \theta \in \Theta_k} x^\top \theta - x_k^\top \theta^*\right) = \sum_{k=1}^{K} x_k^\top (\theta_k - \theta^*) = \sum_{k=1}^{K} x_k^\top \mu_k \ .$$

We now take a peeling-based regret analysis that is quite different from existing analysis techniques:

$$\mathcal{R}^K \leq \sum_{k=1}^{K} x_k^\top (\theta_k - \theta^*) \leq 2^{-L} K + \sum_{\ell=1}^{L} 2^{-\ell+2} \sum_{k=1}^{K} \mathbb{1}\left\{x_k^\top \mu_k \in (2 \cdot 2^{-\ell}, 2 \cdot 2^{-\ell+1}]\right\} ,$$

where $L$ is defined in Algorithm 1. Given $\ell$ and $k$, let $n_{k,\ell}$ be such that $\max_{v:k \leq v \leq K, \ell_v = \ell} |x_k^\top \mu_v| \in (2^{-\ell+n}, 2^{-\ell+n+1}]$ if such $n$ satisfies $n \geq 1$. Otherwise, set $n_{k,\ell} = 0$, which means $\max_{v:k \leq v \leq K, \ell_v = \ell} |x_k^\top \mu_v| \leq 2^{-\ell+n+1}$ with $n = 0$. We then define $G_{\ell,n} := \{s \in [K-1] : \ell_s = \ell, n_{s,\ell} = n\}$ and let $G_{\ell,n}[k] := G_{\ell,n} \cap [k]$. Then,

$$\sum_{k=1}^{K} \mathbb{1}\left\{x_k^\top \mu_k \in (2 \cdot 2^{-\ell}, 2 \cdot 2^{-\ell+1}]\right\} = \sum_{k=1}^{K} \mathbb{1}\{\ell_k = \ell\} \leq 1 + \sum_n \sum_{s \in G_{\ell,n}} 1 \ .$$

Letting $V_{\ell,n,k-1} := 2^{-\ell} I + \sum_{s \in G_{\ell,n}[k-1]} x_s x_s^\top$, a comparison between two matrices $W$ and $V$ is given as follows for every $v \in \{k, \ldots, K\}$:

$$W_{\ell,k-1}(\mu_v) = 2^{-\ell} I + \sum_{s=1}^{k-1} \left(1 \wedge \frac{2^{-\ell}}{|x_s^\top \mu_v|}\right) x_s x_s^\top \geq 2^{-\ell} I + \sum_{s \in G_{\ell,n}[k-1]} \left(1 \wedge \frac{2^{-\ell}}{2^{-\ell+n+1}}\right) x_s x_s^\top$$

$$\geq c \cdot 2^{-n} V_{\ell,n,k-1} \ . \tag{4}$$

For $k \in G_{\ell,n}[K-1]$ and $u = \arg\max_{v:k \le v \le K, \ell_v = \ell} |x_k^\top \mu_v|$, we have

$$2^{-\ell+n} < |x_k \mu_u| \le \|x_k\|_{W_{\ell,k-1}^{-1}(\mu_u)} \|\mu_u\|_{W_{\ell,k-1}(\mu_u)}$$

$$\le \|x_k\|_{W_{\ell,k-1}^{-1}(\mu_u)} \|\mu_u\|_{W_{\ell,u-1}(\mu_u)} \qquad (u \ge k)$$

$$\le c\sqrt{2^n} \|x_k\|_{V_{\ell,n,k-1}^{-1}} \sqrt{2^{-\ell}(\sqrt{A_K \iota} + \iota)} . \qquad ((4) \text{ and Lemma 3(i)})$$

Consequently, $\|x_k\|_{V_{\ell,n,k-1}^{-1}}^2 \ge c\frac{2^{-\ell+n}}{\sqrt{A_K \iota} + \iota}$. Thus, using Lemma 4 with $\tau = 2^{-\ell}$,

$$1 + \sum_{n=0}^{\ell} \sum_{k \in G_{\ell,n}} 1 \le 1 + \sum_n \sum_{k \in G_{\ell,n}} \mathbb{1}\left\{ \|x_k\|_{V_{\ell,n,k-1}^{-1}}^2 \ge c\frac{2^{-\ell+n}}{\sqrt{A_K \iota} + \iota} \right\}$$

$$\le 1 + c \sum_n 2^{\ell-n} (\sqrt{A_K \iota} + \iota) d \ln\left(1 + c4^\ell(\sqrt{A_K \iota} + \iota)\right)$$

$$\le c2^\ell (\sqrt{A_K \iota} + \iota) d \ln\left(1 + c4^\ell(\sqrt{A_K \iota} + \iota)\right) .$$

where we use the fact that $1/\ln(1+q) \le c/q$ for an absolute constant $c$ if $q$ is bounded by an absolute constant. Finally,

$$\sum_{k=1}^K x_k^\top \mu_k \le 2^{-L} K + c \sum_{\ell=1}^L 2^{-\ell} 2^\ell (\sqrt{A_K \iota} + \iota) d \ln\left(1 + c4^\ell(\sqrt{A_K \iota} + \iota)\right)$$

$$= 2^{-L} K + cL(\sqrt{A_K \iota} + \iota) d \ln\left(1 + c4^L(\sqrt{A_K \iota} + \iota)\right) .$$

We choose $L = 1 \vee \lfloor \log_2(K) \rfloor$, which leads to $\mathcal{R}^K \le c\left(\sqrt{A_{K-1} \iota} + \iota\right) d \ln^2\left(1 + cK^2(\sqrt{A_{K-1} \iota} + \iota)\right)$. This concludes the proof.

# 4 Linear Mixture MDP

As linear bandits and linear mixture MDPs have quite a similar nature, we bring the techniques in our analysis of VOFUL2 to improve the regret bound of VARLin of Zhang et al. [31]. A key feature of linear mixture MDP setting is that one can estimate the upper bound of the variance as it is a quadratic function of $\theta^*$ while linear bandits do not have a structural assumption on the variance. Thanks to such a structural property, we obtain a slightly better dependence on the dimension $d$. The confidence set derived for our proposed algorithm is slightly different from that of VARLin as ours is defined with $\forall \mu \in \mathbb{B}_1^d(2)$ rather than an $\epsilon$-net. Our version of VARLin, which we call VARLin2, is described in Algorithm 2. Given $s_h^k$ and $a_h^k$, let us define $P_{s_h^k, a_h^k}(V_{h+1}^k) := \mathbb{E}_{s' \sim P_{s_h^k, a_h^k}}[V_{h+1}^k(s')]$ and $x_{k,h}^m := [P_{s_h^k, a_h^k}^1(V_{h+1}^k)^{2^m}, ..., P_{s_h^k, a_h^k}^d(V_{h+1}^k)^{2^m}]^\top$ and let $L$, $\iota$, and $\delta$ be given as define Algorithm 2. Let $\epsilon_{v,u}^m(\theta) := \theta^\top x_{v,u}^m - (V_{u+1}^v(s_{u+1}^v))^{2^m}$ for $(v, u) \in [K] \times [H]$, $m \in \{0, 1, ..., L\}$ where $L$ is defined in Algorithm 2.

We construct our confidence set as

$$\Theta_k := \bigcap_{m=0}^L \bigcap_{i \in [L]} \bigcap_{\ell \in [L]} \Theta_k^{m,i,\ell} \qquad (5)$$

where we define $\Theta_k^{m,i,\ell}$ below, based on the data collected up to episode $k-1$. First, let

$$\eta_{k,h}^m := \max_{\theta \in \Theta_k} \{\theta^\top x_{k,h}^{m+1} - (\theta^\top x_{k,h}^m)^2\}$$

$$\text{and} \quad \mathcal{T}_{k,h}^{m,i} := \{(v, u) \in ([k] \times [H]) \cup (\{k\} \times [h]) : \eta_{v,u}^m \in (2^{-i}, 2^{1-i}])\} .$$

We naturally define $\mathcal{T}_{k,h}^{m,L+1} := \{(v, u) \in \mathcal{T}_{k,h}^{m,i} : \eta_{v,u}^m \le 2^{-L}\}$. With $\iota$ defined in Algorithm 2, define

$$\Theta_{k-1}^{m,i,\ell} := \left\{ \theta \in \mathbb{B}_1^d(1) : \left| \sum_{(v,u) \in \mathcal{T}_{k-1,H}^{m,i}} \overline{\left((x_{v,u}^m)^\top \mu\right)}_\ell \epsilon_{v,u}^m(\theta) \right| \le \right.$$

**Algorithm 2** VARLin2

1: **Initialize:** $L = \lfloor \log_2 HK \rfloor + 1$, $\iota = 3\ln((2HK)^{2(d+3)}/\delta)$, $\delta \leq e^{-1}$.
2: **for** $k = 1, 2, \ldots, K$ **do**
3:     **for** $h = H, \ldots, 1$ **do**
4:         For each $(s, a) \in \mathcal{S} \times \mathcal{A}$, define $Q_h^k(s, a) = \min\{1, r(s, a) + \max_{\theta \in \Theta_{k-1}} \sum_{i=1}^d \theta_i P_{s,a}^i V_{h+1}^k\}$
        where $\Theta_{k-1}$ is defined in Lemma 5
5:         For each state $s$, $V_h^k(s) = \max_{a \in \mathcal{A}} Q_h^k(s, a)$.
6:     **end for**
7:     **for** $h = 1, \ldots, H$ **do**
8:         Choose $a_h^k = \arg\max_{a \in \mathcal{A}} Q_h^k(s_h^k, a)$.
9:         Observe a reward $r_h^k$ and the next state $s_{h+1}^k$.
10:     **end for**
11: **end for**

$$4\sqrt{\sum_{(v,u) \in \mathcal{T}_{k-1,H}^{m,i}} \overline{\left((x_{v,u}^m)^\top \mu\right)_\ell^2} \eta_{v,u}^m \iota + 4 \cdot 2^{-\ell} \iota}, \forall \mu \in \mathbb{B}_1^d(2) \Bigg\} \tag{6}$$

We show that the confidence set is correct w.h.p. in the following lemma.

**Lemma 5.** *(Confidence set for MDP)* $\mathbb{P}(\forall k \in [K], \theta^* \in \Theta_k) \geq 1 - \delta$.

The consequence is that the $Q$ values computed in VarLin2 is optimistic with high probability due to the following property:

**Lemma 6.** *For every* $k \geq 1$, $\theta^* \in \Theta_k \implies \forall h, s, a : Q_h^k(s, a) \geq Q^*(s, a)$.

Now with the confidence set defined above we state our main result.

**Theorem 2.** *With probability at least 1-$\delta$,*

$$\mathcal{R}^K = \sum_{k=1}^K [V^*(s_1^k) - V^k(s_1^k)] = \tilde{O}(d\sqrt{K \log^2(1/\delta)} + d^2 \log(1/\delta)) .$$

*where* $\tilde{O}$ *hides poly-logarithmic dependence on* $\{d, K, H, \ln(1/\delta)\}$.

## 4.1 Proof of Theorem 2

The main idea of the proof is to infuse a peeling-based argument together with elliptical potential count lemma to both the planning horizon and episode. Noting that the regret of the predicted variance is controlled by the variance of variance, one can expect to reduce the total regret using this information, as done in [31]. We begin by introducing relevant quantities that are parallel with those in linear bandits. Let us first introduce the following lemma.

**Lemma 7.** *Let* $x_1, \ldots, x_T \in \mathbb{R}^d$ *be a sequence of vectors with* $\|x_t\|_2 \leq X$ *for all* $t \in [T]$. *Let* $V_t = \lambda I + \sum_{s=1}^t x_s x_s^\top$. *Let* $0 = \tau_0 < \tau_1 < \tau_2 < \ldots < \tau_z = T$ *where* $\tau_i$ *marks the last time step of the i-th block formed by* $\{\tau_{i-1} + 1, \ldots, \tau_i\}$ *for all* $i \in \{0, \ldots, z\}$. *Let* $\mathsf{anc}(t)$ *be the 'anchor' of* $t$, *the last time step of the i-th block such that the* $(i+1)$*-th block contains* $t$: $\mathsf{anc}(t) = \max\{\tau_i : i \in \{0, \ldots, z\}, \tau_i < t\}$. *Let* $r > 0$. *Define* $J \subseteq [T]$ *to be the set of indices* $t$ *such that* $\|x_t\|_{V_{\mathsf{anc}(t)}^{-1}}^2 > r\|x_t\|_{V_{t-1}^{-1}}^2$ *is true for the first time in the block containing* $t$. *Then,*

$$|J| \leq \frac{2}{\ln(r)} d \ln\left(1 + \frac{2/e}{\ln(r)} \frac{X^2}{\lambda}\right) .$$

To keep track of episode-horizon index pairs concisely, we use a *flat index* $t \in [T]$ where $T := HK$. Specifically, an episode $k$ and a horizon $h$ corresponds to the flat index $t = (k-1)H + h$. Let $\mathsf{t}(k,h) := (k-1)H + h$. Let $\mathsf{k}(t)$ and $\mathsf{h}(t)$ be the mapping from $t$ to its corresponding episode and horizon index respectively so that $k = \mathsf{k}(\mathsf{t}(k,h))$ and $h = \mathsf{h}(\mathsf{t}(k,h))$. By taking $\tau_k$ in Lemma 7 as $\mathsf{t}(k,H)$, we have that $\mathsf{anc}(t) := \mathsf{t}(t-1, H)$. We define $\mathcal{T}_t^{m,i} := \{\mathsf{t}(k',h') : (k',h') \in \mathcal{T}_{k,h}^{m,i}\}$ and $\mu_t^m := \mu_{\mathsf{k}(t),\mathsf{h}(t)}^m$. Similarly, we define $x_t^m$, etc., by replacing the subscript $k, h$ with $t$. Hereafter, any appearance of subscript $k, h$ can be replaced with $t$ such that $t = \mathsf{t}(k,h)$ without changing the meaning.

Given $m$, $k$ and $h$, we define $\ell_{k,h}^m$ as the integer $\ell$ such that $(x_{k,h}^m)^\top \mu_{k,h}^m \in (2 \cdot 2^{-\ell}, 2 \cdot 2^{-\ell+1}]$ where $\mu_{k,h}^m := \theta_{k,h}^m - \theta^*$ and $\theta_t^m = \arg\max_{\theta \in \Theta_{k-1}} |\{\theta^\top x_t^{m+1} - (\theta^\top x_t^m)^2\}|$. For simplicity, we abbreviate $\ell_{k,h}^m$ by $\ell$. Define $W_t^{m,i,\ell}(\mu) := 2^{-\ell}I + \sum_{s \in \mathcal{T}_t^{m,i}} \left(1 \wedge \frac{2^{-\ell}}{|(x_s^m)^\top \mu|}\right) x_s^m (x_s^m)^\top$ and introduce a shorthand $W_{\mathsf{anc}(t)}^{m,i,\ell} := W_{\mathsf{anc}(t)}^{m,i,\ell}(\mu_t^m)$ as before. With the definition above we have the following:

$$2^{-\ell}\|\mu_t^m\|^2 + \sum_{s \in \mathcal{T}_{\mathsf{anc}(t)}^{m,i}} \overline{\left((x_s^m)^\top \mu_t^m\right)}_\ell (x_s^m)^\top \mu_t^m = \|\mu_t^m\|_{W_{\mathsf{anc}(t)}^{m,i,\ell}}^2 \ .$$

We now show the key result of the confidence set of VarLin2 that parallels Lemma 3 for bandits.

**Lemma 8.** *Fix $m \in \{0, \ldots, L_\mathsf{m}\}$ and $i \in [L]$. Let $t \in \mathcal{T}_T^{m,i}$. Then, with $\ell = \ell_t^m$,*

$$\|\mu_t^m\|_{W_{\mathsf{anc}(t)}^{m,i,\ell}}^2 \leq c_M \cdot \max\{\sqrt{2^{-i}\iota}, \sqrt{2^{-\ell}\iota}\}$$

*where $c_M > 0$ is an absolute constant.*

What is different from the linear bandit problem is that we do not update $\theta$ until the planning horizon is over and an additional layer for peeling is imposed on variance. In [31], the authors introduce an indicator $I_h^k$ to characterize episode-horizon pairs for which growth of the norm of $\mu$ with respect to $W_t$ is controlled by the norm with respect to $W_{\mathsf{anc}(t)}$ with $d^2$ growth rate, i.e.,

$$I_h^k := \mathbb{1}\{\|\mu_t\|_{W_t^{m,i,\ell}} \leq 4(d+2)^2 \|\mu_t\|_{W_{\mathsf{anc}(t)}^{m,i,\ell}}\} \ .$$

where $t = \mathsf{t}(k, h)$. Our novelty novelty lies in being able to replace $4(d+2)^2$ above by a constant rate $r$ that is set to 2 later (modulo some differences due to technical reasons). To distinguish we denote such a set by $I_{k,h}$. See B.5 for the definition of $I_{k,h}$ and the proof of the following lemma.

**Lemma 9.** $\sum_{k=1}^K \sum_{h=1}^{H-1} I_{k,h} - I_{k,h+1} \leq O(\frac{d}{\ln(r)}) \log(dHK(1 + d^2/\ln(r)))$ *for $r > 0$.*

Note here that once we fix $r$ such as $r = 2$, the bound can be replaced by $O(d \log^5(dHK))$. We now use the following regret decomposition due to [31] which just come from replacing $I_h^k$ by $I_{k,h}$.

**Lemma 10.** *(Zhang et al. [31])* $\mathcal{R}^K \leq Reg_1 + Reg_2 + Reg_3 + \sum_{k=1}^K \sum_h^{H-1}(I_{k,h} - I_{k,h+1})$ *where $Reg_1 = \sum_{k,h}(P_{s_h^k,a_h^k}V_{h+1}^k - V_{h+1}^k(s_{h+1}^k))I_{k,h}$, $Reg_2 = \sum_{k,h}(V_h^k(s_h^k) - r_h^k - P_{s_h^k,a_h^k}V_{h+1}^k)I_{k,h}$, and $Reg_3 = \sum_{k=1}^K(\sum_{h=1}^H r_h^k - V_1^{\pi_k}(s_1^k))$.*

Let $\breve{x}_{k,h}^m := x_{k,h}^m I_{k,h}$ and define $R_m$, $M_m$ as

$$R_m := \sum_{k,h}(\breve{x}_{k,h}^m)^\top \mu_{k,h}^m, \quad \text{and} \quad M_m := \sum_{k,h}(P_{s_h^k,a_h^k}(V_{h+1}^k)^{2^m} - (V_{h+1}^k(s_{h+1}^k))^{2^m})I_{k,h}.$$

We have that $Reg_1 = M_0$ and $Reg_2 \leq R_0$ since

$$Q_h^k(s,a) - r(s,a) - P_{s,a}V_{h+1}^k \leq \max_{\theta \in \Theta_k} x_{k,h}^0(\theta - \theta^*).$$

To proceed, we first note that $\sum_{k,h}(I_{k,h} - I_{k,h+1})$ and $Reg_3$ are bounded by $O(d \log^5(dHK))$ and $O(\sqrt{K \log(1/\delta)})$ respectively from Lemma 9 and Lemma 16. Since $Reg_1 + Reg_2 \leq R_0 + M_0$, it remains to find a bound on $R_0 + M_0$. This, however, involves solving a series of recursive inequalities. We leave the details in the appendix and provide a high-level description below.

Let us begin with Lemma 15 in the appendix that shows

$$|M_m| \leq \tilde{O}(\sqrt{M_{m+1} + d + 2^{m+1}(K+R_0)\log(1/\delta)} + \log(1/\delta)) \tag{7}$$

where the RHS is a function of $\sqrt{M_{m+1}}$ and $\sqrt{R_0}$. Taking Proposition 1 below for granted and combining it with the relation from Zhang et al. [31, Eq. (57)] showing

$$\sum_{k,h}\eta_{k,h}^m I_{k,h} \leq M_{m+1} + O(d \log^5(dHK)) + 2^{m+1}(K + R_0)) + R_{m+1} + 2R_m,$$

one arrives at $R_m \leq \tilde{O}(d^{1/2}\sqrt{(M_{m+1}+2^{m+1}(K+R_0)+R_{m+1}+2R_m+d})$. This bound is the key improvement we obtain via our peeling-based regret analysis. Specifically, the bound on $R_m$ obtained by [31] has $d^4$ and $d^6$ in place of $d^{1/2}$ and $d$ above.

We first show how our regret bound helps in obtaining the stated regret bound and then present Proposition 1. Noting that both $R_L$ and $M_L$ are trivially bounded by $HK$, one can solve the series of inequalities on $R_m$ and $|M_m|$ to obtain a bound on $R_0$:

$$R_0 \leq \tilde{O}\big(d^2 \log(1/\delta) + \sqrt{d^2(K+R_0)\log(1/\delta)}\big). \tag{8}$$

Solving it for $R_0$, we obtain $R_0 \leq \tilde{O}\big(d^2 \log(1/\delta) + \sqrt{d^2 K \log(1/\delta)}\big)$. One can now plug in $R_0$ to the bound (7) and obtain a bound on $|M_0|$ in a similar way as follows, which concludes the proof:

$$|M_0| \leq \tilde{O}(d\sqrt{K\log^2(1/\delta)} + d^2 \log(1/\delta)).$$

We now show the key proposition that allows us to improve the bound on $R_m$. In the paper by Zhang et al. [31], $d^4$ was derived while we propose the following.

**Proposition 1.** *Let $\breve{\eta}_{k,h}^m := \eta_{k,h}^m I_{k,h}$. Then, we have*

$$R_m \leq O(d^{0.5}\log^{2.5}(HK)\sqrt{(1 + \textstyle\sum_{k,h}\breve{\eta}_{k,h}^m)\iota\log(d\iota)} + d\log^3(HK)\iota\log(d\iota))$$

*Proof.* Define $\mathcal{T}^{m,i,\ell} := \{t \in \mathcal{T}_T^{m,i} : (x_t^m)^\top \mu_t^m \in (2\cdot 2^{-\ell}, 2\cdot 2^{1-\ell}]\}$ and split the time steps $\mathcal{T}^{m,i,\ell}$ by

$$\mathcal{T}^{m,i,\ell,\langle 1\rangle} := \left\{t \in \mathcal{T}^{m,i,\ell} : \left\|\mu_{k,h}^m\right\|_{W_{\mathrm{anc}(a)}^{m,i,\ell}} \leq c_M\sqrt{2^{-i}\iota}\right\} \quad \text{and} \quad \mathcal{T}^{m,i,\ell,\langle 2\rangle} := \mathcal{T}^{m,i,\ell} \smallsetminus \mathcal{T}^{m,i,\ell,\langle 1\rangle}.$$

Having defined $I_{k,h}$ with $r = 2$, we also denote $\breve{\mathcal{T}}^{m,i,\ell,\langle z\rangle} := \mathcal{T}^{m,i,\ell,\langle z\rangle} \cap \{t \in [T] : I_t = 1\}$. Now we decompose $R_m$ as

$$R_m = \sum_{t \in [T]}(\breve{x}_t^m)^\top \mu_t^m = \sum_{i,\ell}\sum_{t \in \breve{\mathcal{T}}^{m,i,\ell,\langle 1\rangle}}(\breve{x}_t^m)^\top \mu_t^m + \sum_{i,\ell}\sum_{t \in \breve{\mathcal{T}}^{m,i,\ell,\langle 2\rangle}}(\breve{x}_t^m)^\top \mu_t^m.$$

Fix $m$, $i$ and $\ell$ and focus on $\sum_{t \in \breve{\mathcal{T}}^{m,i,\ell,\langle z\rangle}}(\breve{x}_t^m)^\top \mu_t^m$ for $z = 1, 2$. Hereafter, we omit the superscripts and subscripts of $(m, i, \ell)$ to avoid clutter, unless there is a need. Note that for $t \in \breve{\mathcal{T}}^{\langle 1\rangle,n}$ and $b$ such that $t < b \in \mathcal{T}^{\langle 1\rangle}$,

$$W_{\mathrm{anc}(t)}(\mu_b) = 2^{-\ell}I + \sum_{t' \in \mathcal{T}_{\mathrm{anc}(t)}^{m,i,\ell}}(1 \wedge \frac{2^{-\ell}}{|x_{t'}^\top \mu_b|})x_{t'}x_{t'}^\top \geq 2^{-\ell}I + \sum_{t' \in \mathcal{T}_{\mathrm{anc}(t)}^{\langle 1\rangle,n}}(1 \wedge \frac{2^{-\ell}}{2^{-\ell+n+1}})x_{t'}x_{t'}^\top$$

$$\geq 2^{-\ell}I + 2^{-n-1}\sum_{t' \in \mathcal{T}_{\mathrm{anc}(t)}^{\langle 1\rangle,n}}x_{t'}x_{t'}^\top$$

$$\geq c2^{-n}V_{\mathrm{anc}(t)}^{\langle 1\rangle,n}.$$

For the same $t$, letting $b = \arg\max_{t \leq b' \in \breve{\mathcal{T}}^{\langle 1\rangle}}|x_t^\top \mu_{b'}|$,

$$2^{-\ell+n} \leq |x_t^\top \mu_b| \leq \|x_t\|_{W_{\mathrm{anc}(t)}^{-1}(\mu_b)}\|\mu_b\|_{W_{\mathrm{anc}(t)}(\mu_b)} \leq \sqrt{2^n}\|x_t\|_{(V_{\mathrm{anc}(t)}^{\langle 1\rangle,n})^{-1}}\|\mu_b\|_{W_{\mathrm{anc}(b)}(\mu_b)}$$

$$\leq c\sqrt{r}\sqrt{2^n}\|x_t\|_{(V_{t-1}^{\langle 1\rangle,n})^{-1}}\sqrt{2^{-i}\iota} \qquad \text{(by } b \in \breve{\mathcal{T}}^{\langle 1\rangle}\text{)}$$

This implies that $\|x_t\|_{(V_{t-1}^{\langle 1\rangle,n})^{-1}}^2 \geq c\frac{2^{-2\ell+n}}{r2^{-i}\iota}$. Thus,

$$\sum_{t \in \breve{\mathcal{T}}^{\langle 1\rangle}}x_t^\top \mu_t \leq c2^{-\ell}\sum_{t \in \breve{\mathcal{T}}^{\langle 1\rangle}}1 \leq c2^{-\ell}\sqrt{|\breve{\mathcal{T}}^{\langle 1\rangle}|\sum_{t \in \breve{\mathcal{T}}^{\langle 1\rangle}}1} \leq c2^{-\ell}\sqrt{|\breve{\mathcal{T}}^{\langle 1\rangle}|\sum_{n=0}^{\ell}\sum_{t \in \breve{\mathcal{T}}^{\langle 1\rangle,n}}1}$$

$$\leq c2^{-\ell}\sqrt{|\breve{\mathcal{T}}^{\langle 1\rangle}|\sum_{n=0}^{\ell}\sum_{t \in \breve{\mathcal{T}}^{\langle 1\rangle,n}}\mathbb{1}\left\{\|x_t\|_{(V_{t-1}^{\langle 1\rangle,n})^{-1}}^2 \geq c\frac{2^{-2\ell+n}}{r2^{-i}\iota}\right\}}$$

$$\leq c2^{-\ell}\sqrt{|\breve{\mathcal{T}}^{\langle 1\rangle}|\sum_{n=0}^{\ell}\sum_{t \in \mathcal{T}^{\langle 1\rangle,n}}\mathbb{1}\left\{\|x_t\|_{(V_{t-1}^{\langle 1\rangle,n})^{-1}}^2 \geq c\frac{2^{-2\ell+n}}{r2^{-i}\iota}\right\}}$$

$$(\breve{\mathcal{T}}^{\langle 1\rangle,n} \subseteq \mathcal{T}^{\langle 1\rangle,n})$$

$$\leq c2^{-\ell}\sqrt{|\breve{\mathcal{T}}^{\langle 1\rangle}|\sum_{n=0}^{\ell}\frac{r2^{-i}\iota}{2^{-2\ell+n}}d\ln\left(1+c\frac{r2^{-i}\iota}{2^{-2\ell+n}2^{-\ell}}\right)}$$

$$\leq c\sqrt{dr|\breve{\mathcal{T}}^{\langle 1\rangle}|2^{-i}\iota\ln\left(1+cr\iota 8^{\ell}\right)}\ \leq c\sqrt{dr(1+\sum_{t\in\breve{\mathcal{T}}^{\langle 1\rangle}}\eta_t^m)\iota\ln\left(1+cr\iota 8^{\ell}\right)}$$

where we use the fact that $|\mathcal{T}^{\langle 1\rangle}|\cdot 2^{-i}\leq O(1+\sum_{t\in\breve{\mathcal{T}}^{\langle 1\rangle}}\eta_t^m)$, which is straightforward by the definition. The summation over $\breve{\mathcal{T}}^{\langle 2\rangle}$ can be handled in a similar way and the details of the proof is provided in Section B.7 in our appendix. $\square$

## 5 Conclusion

In this work, we have made significant improvements in the regret upper bounds for linear bandits and linear mixture MDPs by employing a novel peeling-based regret analysis based on the elliptical potential count lemma. Our study opens up numerous future research directions. First, the optimal regret rates are still not identified for these problems. It would be interesting to close the gap between the upper and lower bound. Second, our algorithms are not computationally tractable. We believe computationally tractable algorithms, even at the price of increased regret, may lead to practical algorithms. Finally, characterizing variance-dependent uncertainty in the linear regression setting without prior knowledge of variances is an interesting statistical problem on its own. Identifying novel estimators for it and proving their optimal coverage would be interesting.

## Acknowledgments and Disclosure of Funding

The authors thank Liyu Chen for finding an error in our earlier version. Insoon Yang is supported in part by the National Research Foundation of Korea (MSIT2020R1C1C1009766), the Information and Communications Technology Planning and Evaluation (IITP) grants (MSIT2022-0-00124, MSIT2022-0-00480), and Samsung Electronics. Kwang-Sung Jun is supported by Data Science Academy and Research Innovation & Impact at University of Arizona.

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
