# OpenReview forum: "Improved Regret Analysis for Variance-Adaptive Linear Bandits and Horizon-Free Linear Mixture MDPs"
_NeurIPS.cc/2022/Conference — NeurIPS 2022 Accept_

### Official Review · Reviewer_8K2h · 2022-07-08

**Rating:** 5
**Confidence:** 3
**Soundness:** 3 good
**Presentation:** 3 good
**Contribution:** 3 good

**Summary:**

This work focused on the linear bandit and linear mixture MDPs. The author proposes a novel variance-dependent algorithm, VOFUL2, for the linear bandit, which improved the regret with a factor of $d^{3}$. For linear mixture MDP, the author also proposes the VARLin 2 algorithm with a horizon-free regret guarantee, which improved the previous results with a factor of $d^{3.5}$.

**Questions:**

My main concern is the following:
1. It seems strange that the dependence of dimension d in linear Mixture MDP is better than the linear bandit problem. For both variance-dependent results in Zhang et al. (2021) and Zhou et al. (2021), the dependence on dimension d in MDP is no worse than the linear bandit problem, and the variance-dependent result can only help to reduce the H-factor with the help of total variance. Is there some problem with the result in linear Mixture MDP, or the result in the linear bandit can be improved? It is better if the author could provide more explanation about this difference. I may increase my score if the author can solve this main concern.

Zhang Z, Yang J, Ji X, et al. Variance-aware confidence set: Variance-dependent bound for linear bandits and horizon-free bound for linear mixture MDP

Zhou D, Gu Q, Szepesvari C. Nearly minimax optimal reinforcement learning for linear mixture Markov decision processes.

Other comments:
1. For the Markov decision process, Zhang et al. (2021) first propose the horizon-free result and show learning MDP is not more complicated than the bandits. It is better if the author can comment on it and compare the result of VARLin 2 with it since both achieve a horizon-free regret guarantee.

Zhang Z, Ji X, Du S. Is reinforcement learning more difficult than bandits? a near-optimal algorithm escaping the curse of the horizon

2. For the RL with linear function approximation, He el. Al 2021 also provides a potential elliptical count and uses the peeling technique to analyze linear MDP and Mixture MDP. Therefore, it is better if the author can comment on it before the Lemma 4 (elliptical potential count) and the peeling-based proof.

He J, Zhou D, Gu Q. Logarithmic regret for reinforcement learning with linear function approximation.

3. For the unit cumulative rewards (Line 9), since the reward $r_h^k$ depends on the random variable $s_h^k,a_h^k$, it is better if the author can define it more clear. For example, should the expected cumulative reward be smaller than 1, almost smaller than 1, or always smaller than 1?

4. In Line 77, the reward $|r_k|\leq 1$ can not implies that $|\epsilon_k|\leq 1$ almostly, since the mean of reward $|r_k|$ is not 0.

5. In the attachment, some comments such as "KJ: add the details to the proof" should be removed.

**Limitations:**

This paper provides theoretical guarantees for learning linear bandit and linear mixture MDP. There is no negative societal impact.

**Strengths And Weaknesses:**

Strength:
1. The VOFUL2 Algorithm improved previous results in unknown-variance linear bandit and reduced the gap to only $\sqrt{d}$.
2. The VARLin2 Algorithm obtains a horizon-free regret guarantee for linear mixture MDP, and the result is near-optimal with the dimension d.

Weakness:
1. The VOFUL2 and VARLin2 algorithms are not novel; they are part following the VOFUL and VARLin 2 algorithms in Zhang et al. (2021).

Zhang Z, Yang J, Ji X, et al. Variance-aware confidence set: Variance-dependent bound for linear bandits and horizon-free bound for linear mixture MDP

2. VOFUL2 and VARLin2 also inherited the disadvantages of computing inefficiently. In addition, without the $\epsilon$-net structure, the VOFUL2 algorithm needs to consider the intersection of the infinite confidence set, which is much more complicated than the original VOFUL algorithm.

3. The Lemma 4 (elliptical potential count) is not new. This Lemma and also the corresponding peeling technique in linear mixture MDPs appears in the proof of Lemmas B.3-B.5 (He el. al 2021) previously.

He J, Zhou D, Gu Q. Logarithmic regret for reinforcement learning with linear function approximation.

---

> ### Author Response · Authors · 2022-08-02
> **[Reviewer 8K2h]**
>
>
> 1. "strange that ... linear mixture MDP vs linear bandit": Please see our common response above. To summarize, there is no contradiction between our linear mixture MDP result and the linear bandit result. We hope it helps, but please let us know if you have other questions.
> 1. "Elliptical potential count and the peeling technique": Indeed, at a high-level, He et al. (2021) apply a similar overall strategy. One immediate difference is that we do not incur an extra dependence on $d$ inside the logarithm, but it could be due to the fact that their lemma is dealing with a different value.
>     * However, our EPC is still novel and tighter than the bound appeared in a concurrent work (e.g., Lemma 6.2 of Wagenmaker et al., "First-Order Regret in Reinforcement Learning with Linear Function Approximation: A Robust Estimation Approach").
>     * The point of view where we introduce the matrix norm w.r.t. $W_{\ell,k-1}(\mu)$ is still novel. This enables the connection to the elliptical potential lemma and the peeling technique. Such a viewpoint is exactly what the paper of VOFUL did not realize, which has made the analysis complicated in our opinion. Indeed, the proof of VOFUL (Zhang et al., 2021) does not use peeling on $x_k^\top \mu_k$!
>     * Our Lemma 7 is still novel and is one of the main contributors to the improved regret bound for the linear mixture MDP.
>     * We added He et al. (2021) and discussion in line 150 in the revised version.
> 1. "the unit cumulative rewards (Line 9)": We assume that the cumulative reward is less than or equal to 1 almost surely for any policy. This is the same assumption as the VOFUL paper. It is possible that we did not understand your question; please feel free to rephrase and ask again.
> 1. "reward $|r_k| \le 1$": It seems that we did not clearly state that $|\theta^*| \leq 1$, which is stated in the main body. If so, $|r_k|\leq 1$ implies $|\epsilon_k| \leq 2 $ rather than 1, as you said. One possible fix is to assume $r_k \in [-1/2,1/2]$ so that $|\epsilon_k| \leq 1$.   We will fix it in the final version.

---

> > ### Author Response · Authors · 2022-08-10
> > **Checking in**
> >
> > Dear reviewer,
> >
> > We wonder if our response has addressed your concern. We are especially curious what the reviewer thinks about our 'no contradiction'. Please let us know if you have further questions; we are more than happy to answer.

---

### Official Review · Reviewer_49qM · 2022-07-10

**Rating:** 6
**Confidence:** 3
**Soundness:** 3 good
**Presentation:** 3 good
**Contribution:** 3 good

**Summary:**

This paper revisits the problem for linear bandits and linear mixture MDPs from Zhang et al. (2021). The authors follow their notion of variance-adaptive regret bound and provide novel analysis to improve such bound. In particular, for the linear bandits setting, the current work improves Zhang et al.’s result by a factor of $O(d^3)$ in the leading regrets. For the linear mixture MDPs, the improvement is a factor of $O(d^{3.5})$. The key of the analysis is a novel peeling-based regret analysis that leverages the elliptical potential ‘count’ lemma, which could be of independent interest to other related problems.

**Questions:**

Please justify or comment on the two weaknesses mentioned in the above Strengths And Weaknesses part. I would like to raise up my score if the authors can solve my concerns.


**Limitations:**

The authors mentioned the limitations of computational issues but does not provide explicit solutions, see my comments in the Strengths And Weaknesses part. And there is no potential negative societal impact of their work.

**Strengths And Weaknesses:**

Strength:
1. The authors made solid theoretical contributions to the literature. The current work significantly improves previous works’ regret bounds, and the techniques are applicable to both linear bandits and linear mixture MDPs. From my perspective, the peeling techniques and the elliptical potential count lemma can be applied to a wider scope beyond the aforementioned linear bandits and linear mixture MDPs.

2. The technical analysis is sound and the writing is easy to follow.

Weakness

1. My main concern is about the practical usage of the current variance-adaptive algorithm. The confidence interval used in Eq. (3) seems quite complicated. I cannot see the clear landscape of this confidence interval. I wonder how to find the action $x_k$ given such a “conceptual” interval. It could be better if the authors could elaborate more on the landscape of the optimization problem using Eq. (3) as the feasible set of $\theta$. For example, the author could explain under what conditions or in what special cases this interval could be efficiently used. Another way is to show some plausible heuristics to calculate line 4 of the VOFUL2, like optimizing $\theta$ and $x_k$ alternately.

2. The current work does not include any empirical evaluations for their algorithm and the contribution is purely theoretical. I guess it is due to the same reason as mentioned above that line 4 of VOFUL2 does not have efficient solutions.

---

> ### Author Response · Authors · 2022-08-02
> **[Reviewer 8K2h]**
>
> We thank the reviewer for acknowledging the implication of our developed analysis tools. As we mentioned in the common response, we believe it is very important to understand that even theoretically the regret bound can be small.
>
> To address your comments,
> 1. Please see our common response for how one can implement our confidence set. For heuristics, here is how we would do.    For a given $\mu$, our confidence set implies that  $|\sum\_{s=1}^k \overline{(x\_s^\top \mu)\_\ell} \epsilon\_s(\theta)| \le 2^{-\ell}\sqrt{2\iota}\sqrt{ \sum\_{s=1}^k \epsilon\_s^2(\theta) + \iota}$. Squaring both sides, it becomes a quadratic constraint on $\theta$.    Now, ideally, one should consider this constraint for every $\mu$ in an $\epsilon$-net, which is not tractable.    So, we would either consider $\mu \in$ {$e_1,..., e_d $} (indicator vectors) or the eigenvectors of $V_k$.    We would also include the confidence set from OFUL.    Altogether, the algorithm becomes a linear programming with $d+1$ quadratic constraints.
> 1. We did explore ways to do some kind of fixed point iterations or alternating optimization, but it was hard to deal with the fact that the confidence set's $\mu$ must capture $\mu_k$, which directly depends on the unknown parameter $\theta^*$.
>
> We can add this discussion in the final version.

---

> > ### Author Response · Authors · 2022-08-10
> > **Checking in**
> >
> > Dear reviewer,
> >
> > We wonder if our response has addressed your concern. Particularly we believe that the weaknesses you mentioned are extremely important and are happy to have a further discussion if you have more questions on those issues.

---

### Official Review · Reviewer_dtVb · 2022-07-11

**Rating:** 7
**Confidence:** 4
**Soundness:** 3 good
**Presentation:** 3 good
**Contribution:** 3 good

**Summary:**

This paper studies variance-adaptive linear bandits and linear mixture MDPs. This paper improves significantly upon regret bounds obtained in the previous work by Zhang et al. (2021). To achieve this, they improved the structure of the confidence set in the previous work by removing the $\epsilon$-net construction in the previous work, and they improved the analyses of the previous work by replacing the complicated volumetric-based argument in the previous work by a simple argument based on their novel elliptical potential "count" lemma.
The improvement on the confidence set and the novel elliptical potential "count" lemma are technically solid.

**Questions:**


1. Why do the main text and the appendix have different bibliography? This looks not common.
2. Line 406-408 looks strange. Did you forget to delete them?
3. Line 411: there is "KJ" in the text. Will it break double-blind?

**Limitations:**

1. The proof roadmap for both linear bandits and linear mixture MDP largely follows upon the previous work, which might limit the novelty of this paper. However, the confidence set  might not be able to become a standalone paper, so I think it inevitable to have this limitation.

**Strengths And Weaknesses:**


Strengths:
1. The regret bound is significantly improved upon the previous work.
2. The analyses is largely simplified upon the previous work, which paves the way for the novel convidence set to be applied to broader theoretical applications.
3. The novel elliptical potential "count" lemma would be of broarder interest to the bandit learning community.
3. The bound for linear mixture MDP is near-optimal up to log factors.

Weakness:
1. The bound for linear bandits is $\sqrt d$ away from being tight.
2. The confidence set is still too complicated to be applied in practice, and optimizing in the confidence set cannot be done in polynomial time.

---

> ### Author Response · Authors · 2022-08-02
> **[Reviewer dtVb]**
>
> We appreciate that you acknowledge the importance of our proofs and results!
> 1. "$\sqrt{d}$ away from being tight": In fact, there is no known argument that could reject the possibility of $d^{1.5}\sqrt{\sum_s^K \sigma_s^2}$ being the optimal; please see the common response and Appendix C of our supplementary material. It is an open problem to identify the optimal variance-adaptive regret bound for linear bandits.
> 1. Thank you for your suggestions. We will merge the bibliography for the final version and cleanup unnecessary comments.

---

### Author Response · Authors · 2022-08-02
**Common response**

We would like to thank the reviewers for their thoughtful comments.
We are grateful to the reviewers for pointing out the strength of our paper.
In particular, reviewer dtVb acknowledged that "The analyses is largely simplified upon the previous work, which paves the way for the novel confidence set to be applied to broader theoretical applications".
Indeed, the original paper's proofs are, in our opinion, rather complicated, long, and unpolished, which we have cleaned up significantly with a more unified point of view that can be summarized by a peeling analysis with the elliptical potential count lemma.
Below, we list our response that needs the attention of more than one reviewer.

1. In fact, we can show that the regret bound of VOFUL2 is $\min \{d\sqrt{K},d^{1.5} \sqrt{\sum_s^K \sigma_s^2} \}$.
The proof can be found in our supplementary material, the last section.
The reason is that the confidence set of VOFUL2 is orderwise never larger than OFUL that has $d\sqrt{K}$ regret bound.
This means that there is no contradiction between our linear mixture MDP result and our linear bandit result.
It's just that if you measure the regret with $\sqrt{K}$, then VOFUL2 has the same rate as OFUL, and it is only when we measure regret with $\sqrt{\sum_s^K \sigma_s^2}$ that we have a factor of $d^{1.5}$.
On a related note, we remark that the optimal variance dependent regret bound is still unknown; it can be anywhere between $d\sqrt{\sum_s^K \sigma_s^2}$ and $d^{1.5}\sqrt{\sum_s^K \sigma_s^2}$.

2. Note that our algorithm can be easily converted into a version where we involve an $\epsilon$-net for each $\theta$ and $\mu$ with the right choice of $\epsilon$ that accounts for the approximation error in the regret bound.
Then, we can still implement it for small enough dimensions.
We predict that $d \le 5$ might be doable if one sufficiently optimizes the code. But of course, this would not be much interesting for practitioners. To get a sense of how to convert the confidence set, in fact, one can see that our confidence set proof already uses an epsilon net anyways.

3. We have made a choice to analyze VOFUL2 rather than VOFUL to avoid clutter (dealing with the approximation error from $\epsilon$-net) in the regret analysis. We mention in passing that the original VOFUL regret proof is, strictly speaking, not correct as they do not take care of the approximation error from $\epsilon$-net. This was raised in the review process (reviewer mxmq therein) and the authors said 'it can be done' and they did not fix it in the final version.

4. We believe our contribution is valuable and worth being published at NeurIPS even if the algorithm is not computationally efficient for the following reason. Before our paper, the optimal variance-dependent regret rate was largely unknown, and no one could reject the possibility that the optimal rate is $d^{4.5} \sqrt{\sum_{k}\sigma_k^2}$. If the optimal rate were indeed this rate, then it would not have been much exciting to develop computationally efficient algorithms because these algorithms would not work well with moderately large $d$'s anyways. Our work rejects such a possibility by showing that the regret bound is almost as good as the case where we know the noise level at each time step, which makes it interesting to develop computationally efficient versions! Indeed, we are currently looking into this.

---

### Meta-Review · Area_Chair_w1yw · 2022-08-23

**Recommendation:** Accept
**Confidence:** Certain

**Metareview:**

This paper gives the first minimax optimal (up to log factors) and horizon-free for linear mixture MDP and improved variance-dependent bound for linear bandits. Furthermore, the paper developed a new peeling-based analysis that can be useful for other problems. These contributions make this paper a strong paper in the theoretical RL community. The AC thus recommends acceptance.

**Award:**

No

---

### Decision · Program_Chairs · 2022-09-14

Accept